# Ships Added Mass Effect on a Flexible Mooring Dolphin in Berthing Manoeuvre

## Aleksander Grm 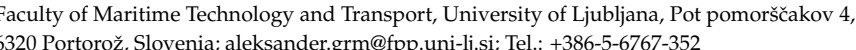

Faculty of Maritime Technology and Transport, University of Ljubljana, Pot pomorščakov 4,
6320 Portorož, Slovenia; aleksander.grm@fpp.uni-lj.si; Tel.: +386-5-6767-352

**Abstract:** This paper deals with the hydrodynamic effect of the ship on a flexible dolphin during a mooring manoeuvre. The hydrodynamic effect refers to the change in momentum of the surrounding fluid, which is defined by the concept of added mass. The main reason for the present study is to answer the question, "What is the effect of the added mass compared to the mass of the ship during the mooring procedure for a particular type of ship?" Measured angular frequencies of dolphin oscillations showed that the mathematical model can be approximated by the zero frequency limit. This simplifies the problem to some extent. The mooring is a pure rocking motion, and the 3D study is approximated by the strip theory approach. Moreover, the calculations were performed with conformal mapping using conformal Lewis mapping for the hull geometry. The fluid flow is assumed to be non-viscous, non-rotating and incompressible. The results showed that the additional mass effect must be taken into account when calculating the flexible dolphin loads.

**Keywords:** added mass; conformal mapping; lewis mapping

## 1. Introduction

Since the beginning of naval history, ships transporting cargo or people from point A to point B have required facilities for safe berthing, loading, and unloading at both points A and B. Over time, ships have grown in size and specialised ships, terminals, and equipment have been built to handle specific types of cargo, such as liquid bulk, dry bulk, and containers. For liquid bulk terminals, a jetty is the typical berthing facility. The ship is usually moored at berths to dedicated breasting dolphins, which may be single-pile flexible dolphins or multi-pile rigid dolphins with fenders.

The primary objective of this work is to estimate the ship added mass. A typical situation of this research geometry and motion is shown in Figures 1 and 2. A ship is moving in a pure sway direction with a constant speed towards the pier. To avoid direct contact with the infrastructure of the liquid cargo terminal, two flexible dolphins reduce the speed of the ship and act as two huge shock absorbers. The current cargo terminal was designed for smaller types of ships, but now larger ships also call at the Port of Koper. As far as safety is concerned, it is also about the safety of the docking process. In the safety analysis of the docking manoeuvre, many different factors need to be analysed in order to get a complete picture of the ship dynamics and the response of the port infrastructure. In this article, we focus exclusively on the estimation of the added mass for such an operation.

Hydrodynamic modelling of added mass phenomena goes way back to names such as Green, Stokes, etc. The influence of added mass has been expressed mathematically and accurately by the expression of the added mass of a sphere. The influence of a free surface on the added mass for surface piercing bodies began many years later. For a given ship, it can be determined by an experimental method. However, the experimental method is limited to a certain condition. To simulate the ship motion, especially in the initial stage of design, the added mass must be calculated by a theoretical method.

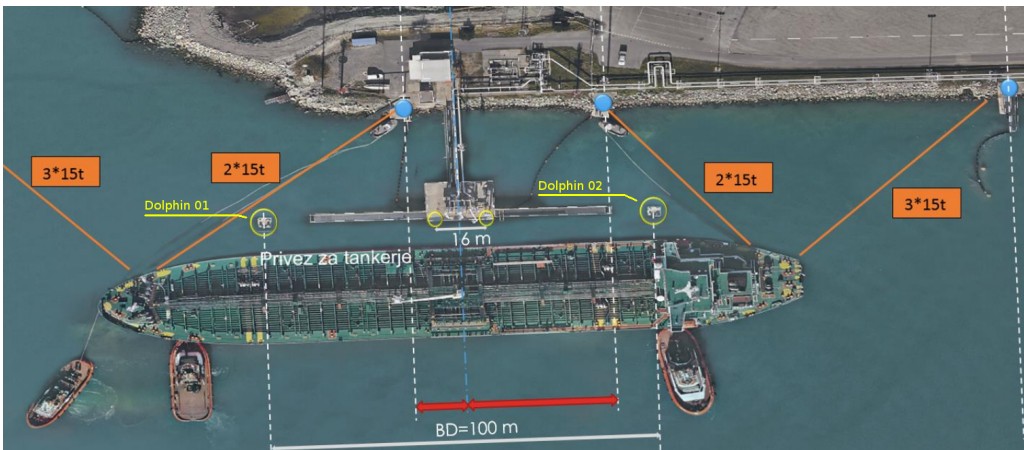

**Figure 1.** View of the berth in Port of Koper. The dolphins are to the right and the left from the central pier-yellow circles on the sea (photo M.Perkovic).

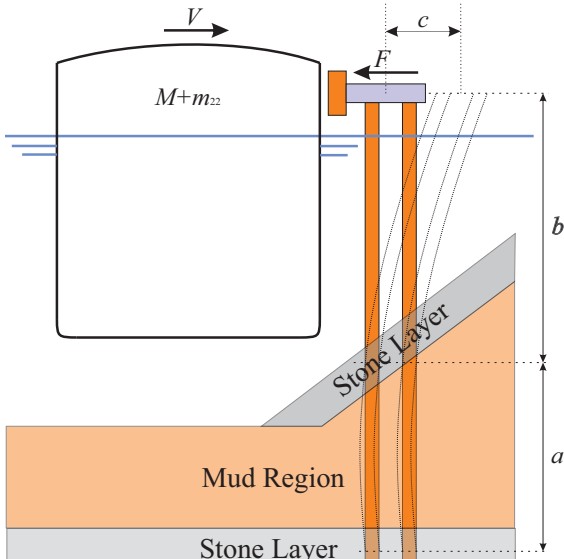

**Figure 2.** Flexible dolphin mooring with all dimensions. The bottom structure consists of different layers of material: Stones and mud. A ship with mass $M + m_{22}$ and velocity $V$ approaches the mooring. The dolphin is curved by $c$ and the force acting at this moment is $F$.

The principal for calculating the added mass for surface piercing bodies began with the work of Ursell [1,2] for a cylindrical cross-section. The mathematical model is based on the multipole expansion approach and is in some sense restricted to simple cross-sectional geometries and infinite water depth. The extension of the model to shallow water goes back to Thorne [3]. An important work by Ursell and co-authors can be found in [4]. The multipole expansion method was later used by many researchers, in particular it is very attractive for those working in theoretical hydrodynamics. The completely different approach began with Frank [5], who developed a method for arbitrary cross-sections based on the integral equation approach. The problem can be solved in the frequency domain, introducing a linear consideration of all quantities involved. However, the mean drift forces of order 2nd can only be obtained with the linear solution, e.g., [6]. In addition, Inglis and Price [7], Newman and Sclavounos [8], and Nakos and Sclavounos [9] are among the most important studies of this type.

All of the above methods implement the potential flow assumption and completely neglect viscous effects. The added mass can typically be approximated as not depending



on viscosity for the particular case of sinusoidal relative motion between the flow and the object [10]. Similarly, viscous effects are negligible for radiated gravity waves due to body motion, but the same is not always true for damping. It is known that viscous damping during roll is typically the most significant viscous effect on the motion of a ship. Lavrov et al. [11] performed CFD calculations using the Navier-Stokes equations with the $k - \omega$ turbulence model to study the flow in the vicinity of 2D ship sections subjected to forced rolling motions. They concluded that for the same shapes, a 10–20% difference in added mass was observed over the entire frequency range compared to results from using a linear frequency domain potential flow code.

The approach taken in the present study is more in line with the Ursell method, combined with the Conformal Mapping approach. Lewis [12] proposed the classical extended Joukowski transformation method, creating the two-parameter Lewis family of ship-like sections. The family was extended by Landweber and Macagno [13,14] to include an additional parameter, the second moment of the cross-sectional area about the horizontal $x$-axis. Ursell's approach was used extensively in ship hydrodynamics by Grim [15], Tasai [16], Porter [17], De Jong [18] and others. Later, Athanassoulis and co-authors [19–21] extended this approach to unsymmetric sections as well. It should be noted that the use of only three parameters leads to a quite satisfactory description of ship sections of conventional hull shapes, as is the case here. This property was exploited, for example, by Grigoropoulos and Loukakis [22,23] to optimize the hull shape in terms of the seakeeping.

The problem of determining added mass traditionally falls within the scope of ship manoeuvrability analysis [24–26]. The manoeuvrability of a ship under various conditions has been studied by several authors [27–31] and many others. The most complex theories of manoeuvring and seakeeping involve nonlinear wave loads with higher-order effects [25]. In our case, it is possible to simplify most of the complex theory. Incoming waves are neglected since the ships sail in mostly closed waters. The measured periods of ship motion are very small [32], so a common approach is to further simplify the motion at a zero frequency limit. In this case, only radiated terms are relevant. A similar approach with experimental setup was also studied in [33,34].

The underlying fluid model is nonviscous, nonrotating, and incompressible to simulate flow around the hull. The ideal flow is represented by a complex velocity potential for the channel geometry (the bottom boundary is included in the geometry—Figure 3). Using the theory of complex functions with conformal mapping, it is possible to solve the flow problem of a complex geometry in a simplified geometry [35–37]. In this study, a cylindrical geometry is mapped to a hull geometry using Lewis mapping [12]. The complex velocity potential is integrated over the simplified geometry to obtain the added mass coefficient. The strip theory approach [38] simplifies the 3D problem to a set of 2D problems. The added mass is calculated for three representative ships: Middle Range oil tanker (MR) with range 25,000 t–55,000 t, Long Range type one oil tanker (LR1) with range 55,000 t–80,000 t and Long Range type 2 oil tanker (LR2) with range 80,000 t–160,000 t. The analysis of the under kill clearance (UKC) effect is also studied. For each type of ship, the velocity field is calculated for 20 different drafts from the summer waterline at the intervals of 0.1 m.

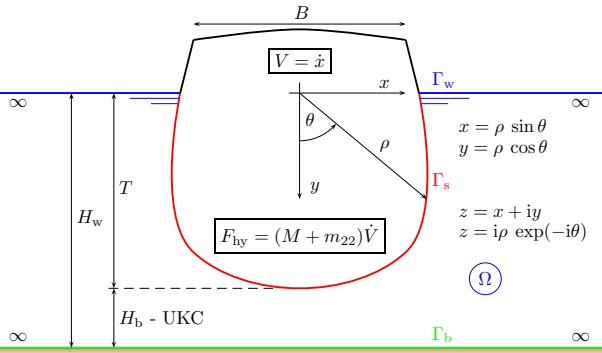

**Figure 3.** Description of the computational domain.

## 2. Formulation of the Problem

The added mass is associated with the change in momentum of the surrounding fluid over time [24]. If the fluid is ideal (non-viscous and irrotational) and incompressible, then the fluid is completely described by the complex velocity potential $\Phi$ in 2D [39]. Consider a two-dimensional ideal and incompressible fluid in a bounded geometry $\Omega$ bounded by the water surface ($\Gamma_w$), the bottom ($\Gamma_b$) and the hull ($\Gamma_s$), as shown in Figure 3. In the $(x, y)$ coordinate system (Figure 3), the velocity potential $\Phi(\boldsymbol{x}, t)$, where $\boldsymbol{x} = (x, y)$ is a point in domain $\Omega$, for a moving body in an otherwise still fluid can be given by the differential equation

$$\frac{\partial^2 \Phi}{\partial x^2} + \frac{\partial^2 \Phi}{\partial y^2} = 0, \quad \boldsymbol{x} \in \Omega \tag{1}$$

and the boundary conditions

$$\frac{\partial \Phi}{\partial y} + k\,\Phi = 0, \qquad \boldsymbol{x} \in \Gamma_w \tag{2a}$$

$$\boldsymbol{n} \cdot \nabla \Phi = 0, \qquad \boldsymbol{x} \in \Gamma_b \tag{2b}$$

$$\boldsymbol{n} \cdot \nabla \Phi = \boldsymbol{n} \cdot \boldsymbol{V}, \qquad \boldsymbol{x} \in \Gamma_s \tag{2c}$$

where $\boldsymbol{n}$ is the normal unit vector always pointing out of domain $\Omega$ and $k$ is a wavenumber defined by the relation $k = \omega^2/g$ (infinite depth [26]), where $\omega$ is the frequency of the oscillating body, $g$ is the acceleration due to gravity, and $\boldsymbol{V}$ is the velocity of the body. Furthermore, the velocity potential for the oscillatory phenomena can be written in the form

$$\Phi(\boldsymbol{x}, t) = \Re\left(\phi(\boldsymbol{x})e^{-\mathrm{i}\omega t}\right), \tag{3}$$

where the potential $\Phi$ is split into the temporal ($e^{-\mathrm{i}\omega t}$) and spatial components ($\phi(\boldsymbol{x})$). It can be shown that the system (1)–(2) is also valid for $\phi$ [26].

In the case we study, the oscillations are very slow ($\omega \ll 1$), so the boundary condition (2a) can be simplified to

$$\frac{\partial \Phi}{\partial y} = \boldsymbol{n} \cdot \nabla \Phi = 0, \quad \boldsymbol{x} \in \Gamma_w \; (\text{for } k \to 0). \tag{4}$$

Now, the solution $\phi$ must satisfy the following system

$$\frac{\partial^2 \phi}{\partial x^2} + \frac{\partial^2 \phi}{\partial y^2} = 0, \qquad \boldsymbol{x} \in \Omega \tag{5a}$$

$$\boldsymbol{n} \cdot \nabla \phi = 0, \qquad \boldsymbol{x} \in \Gamma_w, \Gamma_b \tag{5b}$$

$$\boldsymbol{n} \cdot \nabla \phi = \boldsymbol{n} \cdot \boldsymbol{V}, \qquad \boldsymbol{x} \in \Gamma_s \tag{5c}$$

where the boundary notations are shown in Figure 3. The ship moves with velocity $V$ in $x$ (sway) direction according to the orientation of the coordinate system shown in Figure 3. The fluid flow can be represented by the potential $\phi$ as a moving dipole potential for a body described by a cylindrical shape [24].

Let us convert the $(x, y)$ coordinate system into the complex notation

$$z = x + \mathrm{i}y, \quad x, y \in \mathbb{R}, \quad z \in \mathbb{C}. \tag{6}$$

Such a representation simplifies the solution procedure. It is always possible to write the complex velocity potential as the sum of two real-valued functions

$$\Phi(z) = \phi(x, y) + \mathrm{i}\psi(x, y), \tag{7}$$

where we have the fluid velocity defined as the gradient of the real part of the complex potential [26]

$$\boldsymbol{v} := \nabla \Re(\Phi(z)) = \nabla \phi(x, y). \tag{8}$$

The imaginary part of the complex potential $\psi(x, y) = \Im(\Phi(z))$ is known in the literature as streamlines [24].

The motion of the ship in the sway direction can be represented by a moving complex dipole velocity potential oriented in the $x$ direction and defined as

$$\Phi(z) := \frac{A}{z}, \tag{9}$$

where the constant $A$ is the dipole strength that opposes the fluid at the body boundary and satisfies the nonpenetration boundary condition. The constant $A$ has the unit $[\mathrm{m}^3/\mathrm{s}]$ while the potential in a dimensional for has unit $[\mathrm{m}^2/\mathrm{s}]$. The potential (9) does not satisfy the boundary condition at the bottom $\Gamma_\mathrm{b}$ (2b) and must be corrected somehow. The potential correction is done by using dipole images on both sides of the dipole center in $y$ direction at different positions, which are summed up in an infinite series (method of images [37]). The resulting series converges to a new dipole potential

$$\Phi(z) = \frac{A}{2h} \coth\left(\frac{\pi z}{2h}\right), \tag{10}$$

which also satisfies the missing boundary condition at $\Gamma_\mathrm{b}$ (2b), where the distance between the $\Gamma_\mathrm{w}$ and $\Gamma_\mathrm{b}$ equals to $h = H_\mathrm{w}$ (Figure 3).

**Proposition 1.** *The real part of the potential (10) is the solution of the system (5). The constant $A$ is the strength of the dipole potential $\Phi(z)$ and is obtained from the body boundary condition (5c)*

$$\boldsymbol{n} \cdot \boldsymbol{V} = \boldsymbol{n} \cdot \nabla \phi, \quad \boldsymbol{x} \in \Gamma_\mathrm{b}.$$

**Proof.** The proof of the proposition 1 is trivial. We need to start with $\phi = \Re(\Phi)$ and substitute this into the system (5). Since $\Phi(z)$ is holomorphic, its real part automatically satisfies Laplace's equation. Write the flow velocity $\boldsymbol{v} = (v_x, v_y)$, then the constant $A$ follows from the body boundary condition (5c) at the point $\boldsymbol{x} = (1, 0)$ where the velocity equals $\boldsymbol{v} = (V, 0)$ and the constant $A$ equals

$$A = V \frac{(2h)^2}{\pi} \sinh\left(\frac{\pi}{2h}\right)^2, \tag{11}$$

where $h$ is a dimensionless water column height

$$h = \frac{H_\mathrm{w} - T}{T} + 1, \tag{12}$$

where discussed parameters are shown in Figure 3. Velocity is a time-dependent quantity and the potential can be decomposed as

$$\phi = V\tilde{\phi} \quad \rightarrow \quad \frac{\mathrm{d}\phi}{\mathrm{d}t} = \dot{V}\tilde{\phi}, \tag{13}$$

assuming that the velocity $V$ and potential $\tilde{\phi}$ are related to the velocity and potential in the sway direction [26]. $\square$

Typical solutions of Equation (10) for variables $\phi$ and $\psi$ can be seen in Figures 4–9, for $V = 1$ and various $h$ in the case of a cylindrical body geometry. Let us further rewrite the coordinate system into a more natural one for cylindrical geometry. The transformation from Cartesian coordinates $(x, y)$ to polar coordinates $(\rho, \theta)$ with the notation of the complex plane is

$$x = \rho \sin\theta, \quad y = \rho \cos\theta, \quad x, y, \rho, \theta \in \mathbb{R} \tag{14a}$$

$$z = x + \mathrm{i}y = \mathrm{i}\rho \exp(-\mathrm{i}\theta), \quad z \in \mathbb{C} \tag{14b}$$

as can be seen in Figure 3. The geometry of the cylindrical body can be transformed into a shape similar to the ship-like shape using the conformal mapping $w = f(z)$, preserving the shape of the complex velocity potential $\Phi(z)$ [35]. This fact is used to compute the hydrodynamic force in the cylindrical geometry $\Omega_c$ of the flow generated by the ship geometry $\Omega_s$ Figure 10.

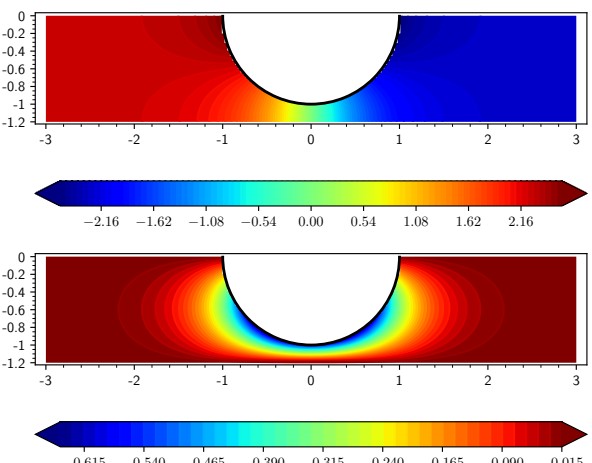

**Figure 4.** Plot of Equation (10) in the form (7). The top plot shows the real part of the complex potential $\phi(x, y)$, the bottom plot shows the imaginary part of the complex potential $\psi(x, y)$ for velocity $V = 1$ and channel gap width $h = 1.2$ for cylindrical geometry $\Gamma_s$ with $\rho = 1$.

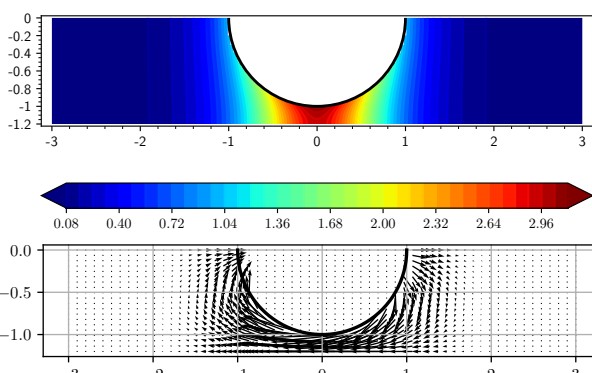

**Figure 5.** Plot of the Equation (8). The top plot shows the velocity amplitude $\|\boldsymbol{v}\|$, the bottom plot shows the velocity vector field for velocity $V = 1$ and the channel gap of width $h = 1.2$ for cylindrical geometry $\Gamma_s$ with $\rho = 1$.

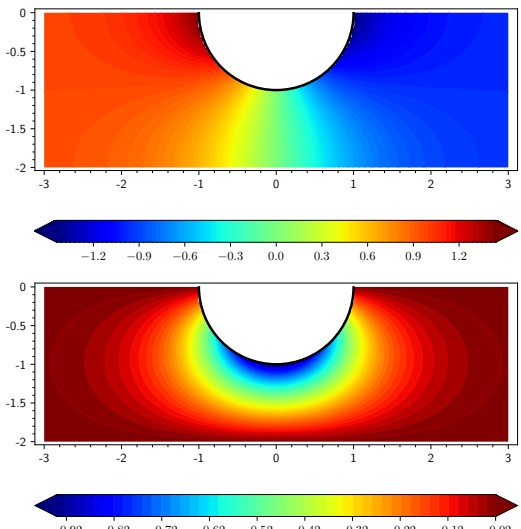

**Figure 6.** Plot of the Equation (10) in the form (7). The top plot shows the real part of the complex potential $\phi(x, y)$, the bottom plot shows the imaginary part of the complex potential $\psi(x, y)$ for velocity $V = 1$ and channel gap width $h = 2.0$ for cylindrical geometry $\Gamma_s$ with $\rho = 1$.

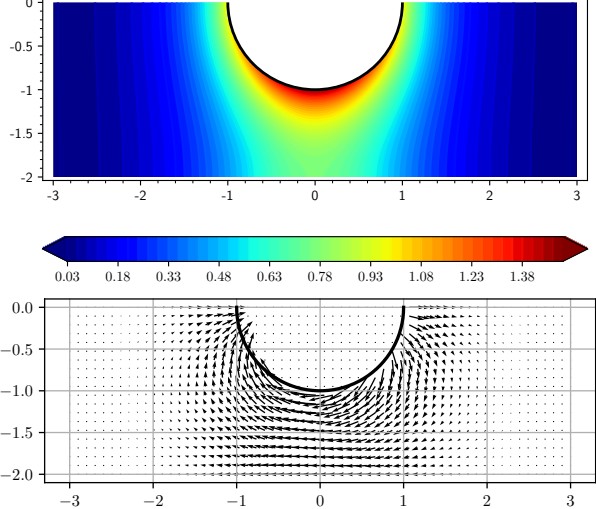

**Figure 7.** Plot of the Equation (8). The top plot shows the velocity amplitude $\|\boldsymbol{v}\|$, the bottom plot shows the velocity vector field for velocity $V = 1$ and the channel gap of width $h = 2.0$ for cylindrical geometry $\Gamma_s$ with $\rho = 1$.

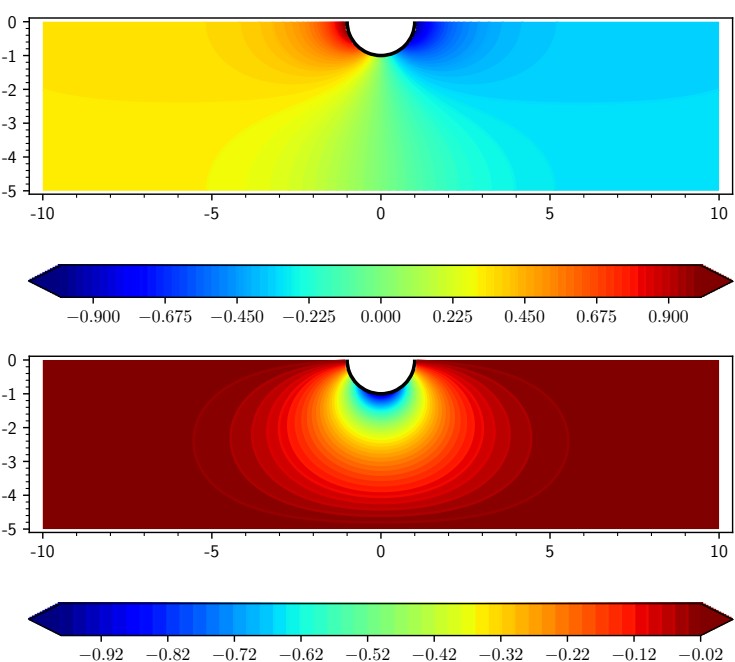

**Figure 8.** Plot of the Equation (10) in the form (7). The top plot shows the real part of the complex potential $\phi(x, y)$, the bottom plot shows the imaginary part of the complex potential $\psi(x, y)$ for velocity $V = 1$ and channel gap width $h = 5.0$ for cylindrical geometry $\Gamma_s$ with $\rho = 1$.

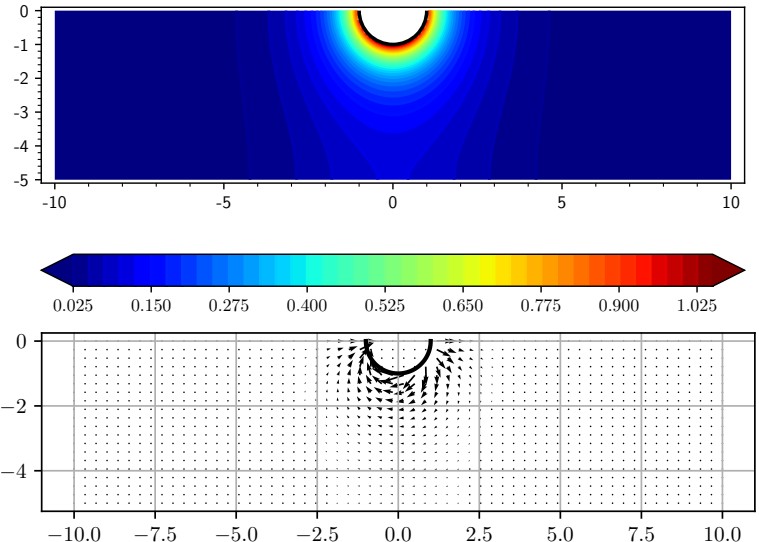

**Figure 9.** Plot of the Equation (8). The top plot shows the velocity amplitude $\|\boldsymbol{v}\|$, the bottom plot shows the velocity vector field for velocity $V = 1$ and the channel gap of width $h = 5.0$ for cylindrical geometry $\Gamma_s$ with $\rho = 1$.

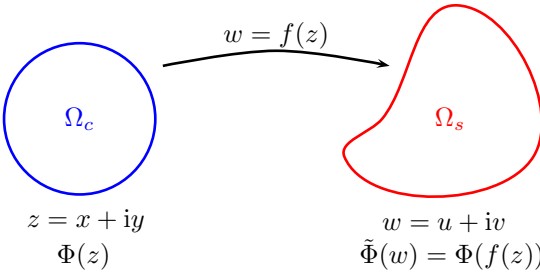

**Figure 10.** Conformal mapping $w = f(z)$ of a circular domain $\Omega_c$ onto a ship-like domain $\Omega_s$ with coordinates and velocity complex potentials preserved by the mapping.

One of the most commonly used conformal mappings for ship-like forms is the Lewis transformation [12], which uses only 3 free parameters $a$, $a_1$ and $a_3$

$$w = a\left(z + \frac{a_1}{z} + \frac{a_3}{z^3}\right), \quad z \in \mathbb{C}, \quad a, a_1, a_3 \in \mathbb{R}, \tag{15}$$

where $a$ only causes the shape to expand/compress, but does not affect the appearance of the shape. The free parameters are determined with the basic parameters of the specific ship cross-section: $B$—maximal breadth, $T$—draft and $S$—area

$$\sigma_s = \frac{S}{BT}, \tag{16a}$$

$$H = \frac{B}{2T}, \tag{16b}$$

$$C_1 = \left(3 + \frac{4\sigma_s}{\pi}\right) + \left(1 - \frac{4\sigma_s}{\pi}\right)\left(\frac{H-1}{H+1}\right)^2, \tag{16c}$$

$$a = \frac{B}{2}(1 + a_1 + a_3), \tag{16d}$$

$$a_1 = (1 + a_3)\frac{H-1}{H+1}, \tag{16e}$$

$$a_3 = \frac{-C_1 + 3 + \sqrt{9 - 2C_1}}{C_1}. \tag{16f}$$

Figure 11 and Table 1 show the data used in the present calculations. The ship constructed from these cross-sections is referred to as the *Lewis ship*. The hydrodynamic properties of sway motion for the Lewis ship are shown in Figure 12 for infinite depth and in Figure 13 for finite depth.

**Table 1.** Lewis mapping coefficients for MR, LR1 and LR2 oil tanker type with $C_b = 0.78$ producing shapes in Figure 11. Only $B_k$ needs to be scaled with $\beta = B/T$ ratio for different draft calculations.

| Section ($k$) | $B_k$ | $T_k$ | $\sigma_k$ | $\tilde{L}_k$ |
|:---:|:---:|:---:|:---:|:---:|
| 0 | 0.8 | 0.2 | 0.60 | 0.05 |
| 1 | 1.2 | 0.9 | 0.50 | 0.05 |
| 2 | 1.6 | 1.0 | 0.68 | 0.05 |
| 3 | 2.0 | 1.0 | 0.93 | 0.05 |
| 4 | 2.0 | 1.0 | 0.99 | 0.60 |
| 5 | 2.0 | 1.0 | 0.93 | 0.05 |
| 6 | 1.8 | 1.0 | 0.68 | 0.05 |
| 7 | 1.2 | 1.0 | 0.56 | 0.05 |
| 8 | 0.3 | 0.7 | 0.56 | 0.05 |

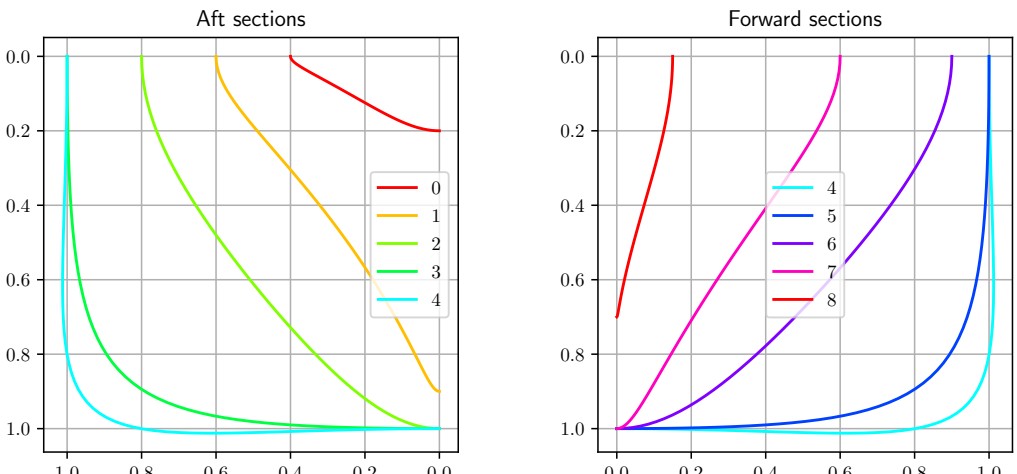

**Figure 11.** Ship cross-sections used in the calculation. The parameters of the cross-section are shown in Table 1.

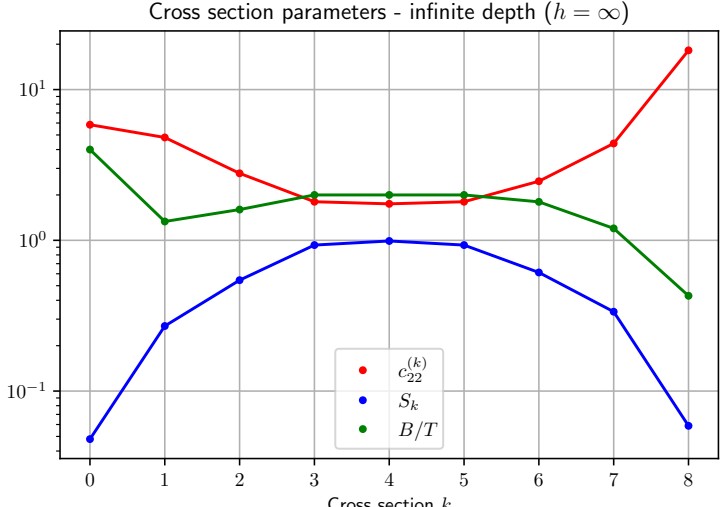

**Figure 12.** Results for Lewis cross-sections $k$ from Table 1 (Figure 11) for infinite water depth. $c_{22}^{(k)}$ is the added mass coefficient, B/T is the ratio of beam/draft cross-section, and $S_k$ is the cross-section area. The scales on the ordinate are logarithmic for better result representation.

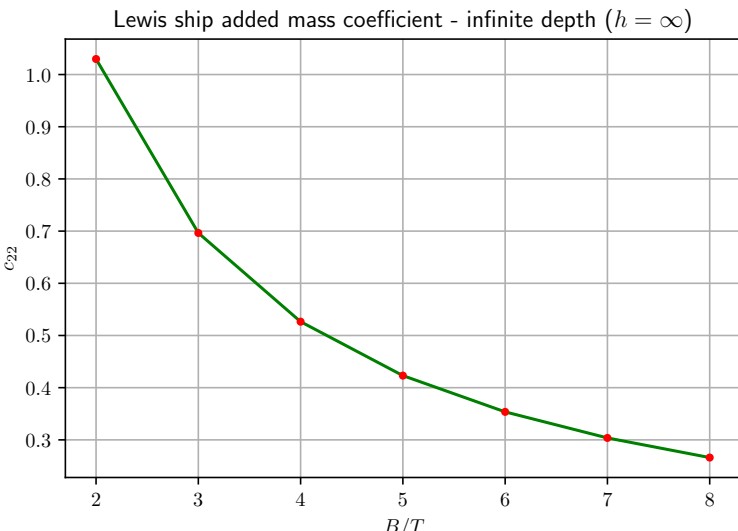

**Figure 13.** Results for Lewis ship defined in Table 1 (Figure 11) for infinite water depth. $c_{22}$ is the added mass coefficient for the Lewis ship, and $B/T$ is the ratio between beam and draft.

The hydrodynamic force resulting from the time variation of the surrounding fluid is defined in [24] and is equal to

$$
\begin{aligned}
\boldsymbol{F} = &-\rho \frac{\mathrm{d}}{\mathrm{d}t} \int_{\Gamma_s} \phi \boldsymbol{n} \, \mathrm{d}S \\
&-\rho \int_{\Gamma_w \cup \Gamma_b} (\boldsymbol{n} \cdot \nabla \phi) \, \nabla \phi \, \mathrm{d}S \\
&+\rho \frac{1}{2} \int_{\Gamma_w \cup \Gamma_b} (\nabla \phi \cdot \nabla \phi) \, \boldsymbol{n}
\end{aligned}
\tag{17}
$$

In the present study, the integrals over the boundary $\Gamma_w \cup \Gamma_b$ are zero, since we are only interested in the sway component of the motion. Splitting the potential $\phi$ into a velocity part and a space part (13) gives the final form of the hydrodynamic force

$$
\begin{aligned}
\boldsymbol{F} &= -\rho \dot{V} \mathcal{V} \int_{\Gamma_s} \tilde{\phi} \boldsymbol{n} \, \mathrm{d}S \\
&= -\dot{V} \rho \mathcal{V} c_{22} \\
&= -\dot{V} m_{22},
\end{aligned}
\tag{18}
$$

where $\mathcal{V}$ is the displacement of the body, $\rho$ is the fluid density, $\dot{V}$ is the acceleration of the body, $c_{22}$ is added mass coefficient in sway mode, and $m_{22}$ is the added mass in sway mode. To calculate the integral over the body surface $\Gamma_s$, we perform the integration for each cross-section $k$ according to Table 1 and add their contribution to the total added mass. The coefficient of the added mass for each cross-section $k$ is calculated in the circular cross-section in space $\Omega_c$ and transferred to the ship cross-section in space $\Omega_s$ using the conformal mapping (15). The integral in (18) is transformed from $\Omega_c$ to $\Omega_s$

$$
c_{22}^{(k)} = \frac{2}{S} \int_0^{\pi/2} \tilde{\phi}(w) \, (\boldsymbol{n}(w) \cdot \boldsymbol{e}_x) \left| \frac{\mathrm{d}w}{\mathrm{d}z} \frac{\mathrm{d}z}{\mathrm{d}\theta} \right| \mathrm{d}\theta,
\tag{19}
$$

where $w = f(z)$ is a conformal mapping (15), $z = z(\rho, \theta)$ is defined in (14b), $\boldsymbol{e}_x$ is a unit vector in $x$ (sway) direction in $\Omega_s$ (Figure 3), and $S$ is the area of the cross-section $k$. The integral (19) is computed as a contour integral over the cylinder in the polar coordinates with radius $\rho = 1$ and $\theta \in [0, \pi/2]$. The term $\mathrm{d}w/\mathrm{d}z$ is the Jacobian of the conformal mapping and $\mathrm{d}z/\mathrm{d}\theta$ follows from the chain rule in the derivative of conformal mapping.

The potential in (19) is written in dimensionless form. The length scale is scaled by $T_i$ (specific draft configuration) and the velocity by the ship velocity $V$ according to the following scheme

$$x = \tilde{x} T_i, \qquad\qquad y = \tilde{y} T_i, \tag{20a}$$
$$\dot{x} = \tilde{v}_x V, \qquad\qquad \dot{y} = \tilde{v}_y V, \tag{20b}$$
$$\ddot{x} = \tilde{a}_x V^2 / T_i, \qquad\qquad \ddot{y} = \tilde{a}_y V^2 / T_i. \tag{20c}$$

The added mass of a cross-section $k$ is given by the cross-section $k$ added mass coefficient (19) multiplied by the respective water density $\rho$ and volume $V_k$

$$m_{22}^{(k)} = c_{22}^{(k)} \rho V_k = c_{22}^{(k)} \rho \left( \tilde{S}_k B T_i \right) \left( \tilde{L}_k L \right), \tag{21}$$

where $\tilde{S}_k$ is the dimensionless cross-sectional area and $\tilde{L}_k$ is the dimensionless cross-sectional length. For each cross-section $k$ the values for $\tilde{S}_k$ and $\tilde{L}_k$ are taken from Table 1, and for each ship type, the constants $B$, $T_i$ and $L$ are taken from Table 2. The final added mass of the ship for the slow sway motion is the sum of all added mass contributions of the cross-sections $k$

$$m_{22} = \sum_{k=0}^{8} m_{22}^{(k)}. \tag{22}$$

Detailed description of added mass calculation procedure is described in next section.

**Table 2.** Oil tanker types used in simulation: $L = L_{bp}$—length between perpendiculars, $B$—maximal breadth, $T_{max}$—draft at summer line, $T_{min}$—minimal draft in simulation, $C_b$ block coefficient. Specific draft $T_i$ is in the interval $[T_{min}, T_{max}]$.

| Type | $L$ [m] | $B$ [m] | $T_{min}$ [m] | $T_{max}$ [m] | $C_b$ |
|------|---------|---------|---------------|---------------|-------|
| MR   | 185.0   | 29.1    | 8.50          | 10.50         | 0.78  |
| LR1  | 220.0   | 36.3    | 10.50         | 12.50         | 0.78  |
| LR2  | 238.0   | 41.3    | 12.20         | 14.20         | 0.78  |

## 3. Results

The ship moves at a relatively slow speed when docked. In this study, the problem's formulation contains many reasonable simplifications to obtain results based only on symbolic derivations. The further simplification of the full 3D problem is based on the strip theory approach. The first step was to decompose the representative geometry of the oil tanker into some cross-sections to obtain relevant shape differences. The Levis map (15) is used to describe different cross-sections. The generated data for each cross-section describing the shape of the oil tanker are shown in Table 1. The results can be seen in Figure 11.

In Figures 4–9 are plots of the complex dipole potential (10) for different values of the water height $h$, where $\rho = 1$ and $V = 1$. The sequence of images for different $h$ shows the difference between the deep water solution ($h \gg 1$) and the shallow water solution ($(h - 1) < 1$. The gap effect can be well observed from the intensity of the velocity potential $\phi$. The maximum value is in the range from 2.5 to 1.2, for water heights from $h = 1.2$ to $h = 5.0$. The magnitude of the velocity in the gap increases with smaller $h$. The higher values of $\phi$ at the cylinder boudary result in a larger additional mass. The magnitude of the velocity in the gap is related to the viscous damping. The larger the magnitude of the velocity in the gap, the smaller the gap width and the stronger the viscous forces act.

Three different representative oil tanker types are studied for the selection of ship types. The different types show the difference in the added mass in terms of ship size, their particulars, and UKC distance. The influence of UKC on the added mass was determined with 20 different ship drafts $T_i$. In this case, the number of draft subdivisions is not a

limit, since the calculations for a single geometry are very fast (order of magnitude of a few seconds). Table 2 gives the main specifications for the different oil tanker types used in the simulation. All three types have the same block coefficient $C_b = 0.78$ with the cross-sectional shapes defined in Table 1 and their particulars defined in Table 2.

To obtain the Lewis cross-sectional forms for various drafts $T_i$, we only need to multiply the coefficient $B_k$ in Table 1 by the constant $\beta_i$

$$B_k \rightarrow \beta_i B_k, \quad \beta_i = \frac{B/T_i}{2}. \tag{23}$$

The ratio $\beta_i$ is defined as the ratio between the ship's beam $B$ and the current ship's draft $T_i$ and the constant ratio $B/T = 2$ for the Lewis ship ( Table 1). The values of a given ship configuration "$i$" ($B/T_i$) are calculated from Table 2. The cross-sectional area $S_k$ for a given configuration $i$ is determined as

$$S_k = \sigma_k \, \beta_i \, B_k \, T_k,$$

where $k = \{0, 1, \ldots, 8\}$ is the cross-section number and $i = \{1, 2, 3, 4, \ldots, N\}$ is the specific draft configuration, where $N$ is the number of different draft scenarios for a given tanker type. In the present case, $N$ was set to 20 to get nice continuous plots. The calculations are very fast, and it takes about SI 1s to calculate a single draft configuration. One of the main considerations in the present work was also the speed of the computation, and it could only be achieved with a semi-analytical approach.

In the previous section, a complete model for calculating the added mass in slow sway motion was formulated. The model is based on a potential flow theory with linear boundary conditions (5). For simple geometries, such as the circular one, the solution $\phi$ of (5) is a pulsating dipole with origin at the free surface (10) with constant $A$ defined in Equation (11). The solution (10) satisfies the PDE system (5) only for a circular body geometry. The added mass coefficient $c_{22}$ of a circular geometry can be easily obtained using the integral (18) for different water heights $h$. Figure 14 shows the solution for the added mass coefficient as a function of different dimensionless gap widths (UKC/$R$). For this particular case, one obtains the explicit expression for the added mass coefficient

$$\begin{aligned}
c_{22}(h) &= \frac{2}{S} \int_0^{\pi/2} \tilde{\phi} \sin \theta \, d\theta \\
&= \frac{4}{\pi} \int_0^{\pi/2} \Re \left[ \frac{2h}{\pi} \sinh^2 \left( \frac{\pi}{2h} \right) \coth \left( \frac{\pi}{2h} z \right) \right] \sin \theta \, d\theta,
\end{aligned} \tag{24}$$

$$z \rightarrow i \exp(-i\theta)$$

$$c_{22}(h) \approx \left[ \frac{1}{3} + \left( \frac{2h}{\pi} \right)^2 \right] \sinh^2 \frac{\pi}{2h}, \quad h > 1, \tag{25}$$

$$h = 1 + \text{UKC}/R,$$

where the term $\coth(x)$ in the integral function (24) has been expanded into Taylor series (see [40]). For $|z| = 1$ the series converges very quickly. Already the first three terms yield the solution error below $10^{-3}$. To obtain the added mass coefficient, the value of the integral must be divided by the area of the cross-section. In this particular case for a circular cross-section with unit radius, the value of the area is $S = (\pi R^2)/2 = \pi/2$. The result shown in Figure 14 will be used later when verifying the results of the proposed method for calculating the added mass of a tanker-type ship.

The average water depth at the liquid terminal in the Port of Koper is approximately $H_w = 14.5$ m. The variable $h$ is calculated using Equation (12) for different Lewis shapes (Table 1) and ship particulars (Table 2) for each draft configuration $T_i$. If $h$ is known, the

coefficient of added mass coefficient $c_{22}$, as defined in Equation (19), can be calculated for each cross-section $k$

$$c_{22}^{(k)} = \frac{2}{S} \int_0^{\pi/2} \tilde{\phi}(w) \, (\boldsymbol{n}(w) \cdot \boldsymbol{e}_x) \left| \frac{\mathrm{d}w}{\mathrm{d}z} \frac{\mathrm{d}z}{\mathrm{d}\theta} \right| \mathrm{d}\theta.$$

Now each term of the integral is explained in detail. Let us begin with the velocity potential

$$\tilde{\phi}(w) = \Re \left[ \frac{2h}{\pi} \, \sinh^2 \frac{\pi}{2h} \, \coth\left( \frac{\pi}{2h} w \right) \right]$$

$$= \Re \left[ \frac{2h}{\pi} \, \sinh^2 \frac{\pi}{2h} \, \coth\left( \frac{\pi}{2h} a \left( z + \frac{a_1}{z} + \frac{a_3}{z^3} \right) \right) \right], \quad z \to \mathrm{i}\exp(-\mathrm{i}\theta),$$

$$= \frac{2h}{\pi} \frac{\sinh^2\left(\frac{\pi}{2h}\right) \sinh\left(\frac{\pi}{2h}\sin\theta\right) \cosh\left(\frac{\pi}{2h}\sin\theta\right)}{\sin^2\left(\frac{\pi}{2h}\cos\theta\right) + \sinh^2\left(\frac{\pi}{2h}\sin\theta\right)}$$

Next is the Jacobian of the transformation

$$\frac{\mathrm{d}w}{\mathrm{d}z} \frac{\mathrm{d}z}{\mathrm{d}\theta} = a[a_1 \exp(\mathrm{i}2\theta) - 3a_3 \exp(\mathrm{i}4\theta) + 1] \exp(-\mathrm{i}\theta)$$

$$= a[a_1 \exp(\mathrm{i}\theta) - 3a_3 \exp(\mathrm{i}3\theta) + \exp(-\mathrm{i}\theta)]$$

$$= a[(a_1 + 1)\cos\theta - 3a_3 \cos 3\theta] + \mathrm{i}a[(a_1 - 1)\sin\theta - 3a_3 \sin 3\theta].$$

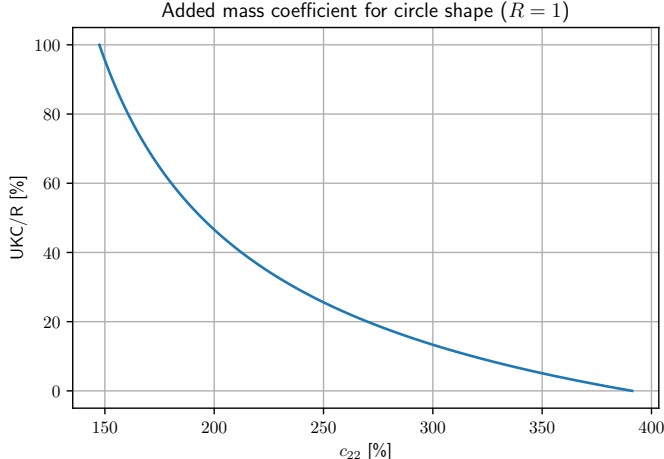

**Figure 14.** Plot of the solution (25) for the added mass coefficient $c_{22}$ with respect to the dimensionless UKC for the solution with circular (Figures 4–9). For the larger UKC, the typical result for the solution with infinite depth ($c_{22} = 100\%$) can be seen. UKC is scaled in dimensionless form with the radius of the circle $R$ and is related to $h$ defined in Equation (25).

The absolute value of the Jacobian is found using the relation $|z| = \sqrt{\Re(z)^2 + \Im(z)^2}$. The normal vector is found by

$$\boldsymbol{r}(w) = \frac{\mathrm{d}w}{\mathrm{d}z}\frac{\mathrm{d}z}{\mathrm{d}\theta} = (r_x, r_y) = (\Re(\boldsymbol{r}), \Im(\boldsymbol{r})),$$

$$\boldsymbol{t}(w) = \frac{\boldsymbol{r}(w)}{|\boldsymbol{r}(w)|} = \frac{(r_x, r_y)}{\sqrt{r_x^2 + r_y^2}},$$

$$\boldsymbol{n}(w) = \mathrm{i}\,\boldsymbol{t}(w),$$
$$n_x = \boldsymbol{n}(w) \cdot \boldsymbol{e}_x = \Re[\boldsymbol{n}(w)],$$
$$n_y = \boldsymbol{n}(w) \cdot \boldsymbol{e}_y = \Im[\boldsymbol{n}(w)],$$

where the vector $\boldsymbol{n}$ is written in complex notation, where the $x$component is equal to the real part ($n_x = \Re(\boldsymbol{n})$) and the $y$component is equal to the imaginary part ($n_y = \Im(\boldsymbol{n})$). The last one is the explanation of the cross-sectional area

$$S = \sigma_k \,\beta_i \,B_k \,T_k,$$

where all the coefficients are taken/calculated form Table 1. The integral is evaluated numerically using the Gaussian quadrature rule for each cross-section $k$ for a single draft and tanker-type configuration with arbitrary accuracy.

The cross-sectional added mass coefficient $c_{22}^{(k)}$ is then multiplied by the corresponding cross-sectional volume to obtain the cross-sectional added mass $m_{22}^{(k)}$ for a given ship type under various draft conditions. Finally, all cross-sectional masses are summed to obtain the ship added mass $m_{22}$ for the sway motion for a given ship type and draft.

Figures 15–17 show the results of calculated ship added mass $m_{22}$ (Equation (22)) for all three tanker types MR, LR1 and LR2 for different drafts. The maximum draft is the draft at the summer load line as given in Table 2. The results show that the added mass increases with ship draft $T$ (green line) resulting in smaller UKC (blue line). Smaller UKC causes higher velocity magnitudes in the hull neighbourhood and higher values of the potential $\phi$ at the ship boundary. The effect appears weakly nonlinear and could not be predicted using crude approximation methods, especially if one is interested in fairly good estimates of the added mass for a given ship type. In contrast, the added mass relative to displacement increases almost linearly (red line). The difference in added mass relative to draft is about 30–45% per 1 meter draft change. Assuming that it is constant over the entire draft range is not good practice in this case, and the effect of draft should always be considered in calculations for the flexible mooring problem.

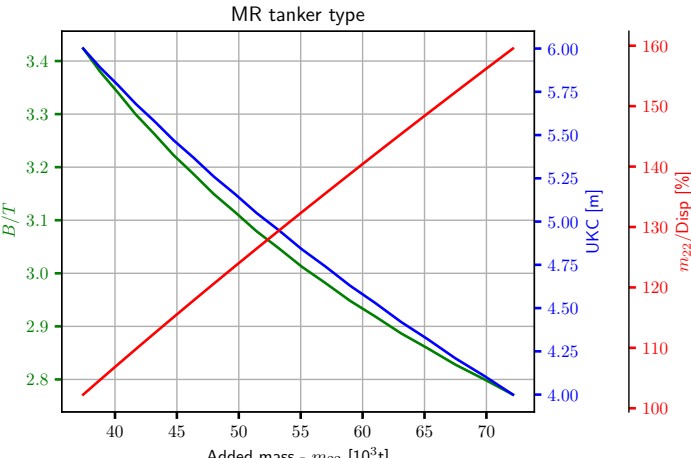

**Figure 15.** Results for the added mass $m_{22}$ of the **MR** tanker type for different drafts. Labeled variables are: $B/T$ (green line—left side scale), UKC (blue line—first right side scale), and the ratio between the added mass and the displacement in percent $m_{22}/Disp$ (red line—second right side scale).

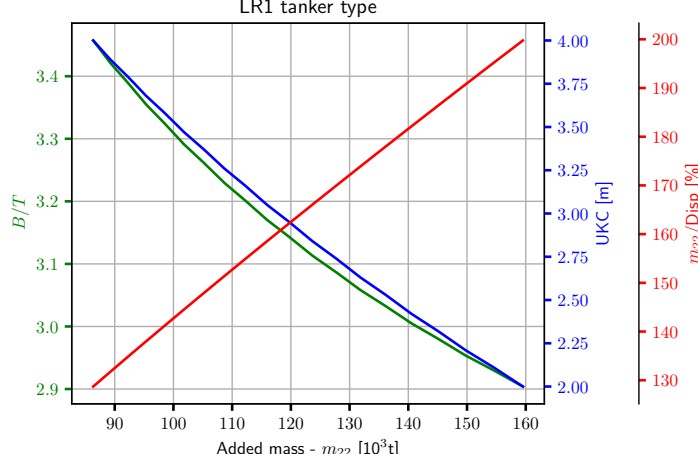

**Figure 16.** Results for the added mass $m_{22}$ of the **LR1** tanker type for different drafts. Labeled variables are: $B/T$ (green line—left side scale), UKC (blue line—first right side scale), and the ratio between the added mass and the displacement in percent $m_{22}/Disp$ (red line—second right side scale).

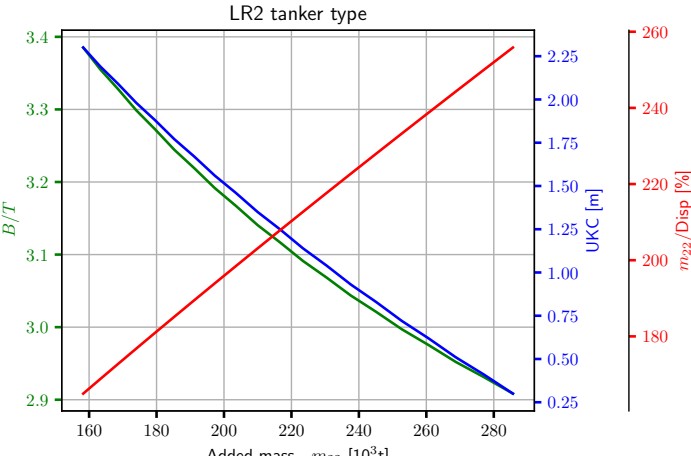

**Figure 17.** Results for the added mass $m_{22}$ of the **LR2** tanker type for different drafts. Labeled variables are: $B/T$ (green line— left side scale), UKC (blue line—first right side scale), and the ratio between the added mass and the displacement in percent $m_{22}/Disp$ (red line—second right side scale).

Figures 18–20 show the same result as in Figures 15–17, but are composed in a different way. Figure 18 shows the added mass as a function of the $B/T$ ratio. The effect of smaller UKC is seen in a faster increase of the added mass. The same phenomenon is observed in Figures 19 and 20. The result shown in Figure 21 is very revealing. The plot shows the added mass coefficient with respect to the dimensionless UKC. Compared with Figure 14 (dash-dot line), the same trend is observed. There is a difference in the added mass coefficient $c_{22}$ between the circular cross-section and the ship-shaped geometry. The difference is due to the different cross-section shapes. Figure 22 is from Vugts research published in [33] and clearly shows the dependence on the $B/T$ ratio with respect to the added mass coefficient $c_{22}$ for the square cross-section. The larger the $B/T$ ratio is, the smaller the added coefficient is. In our case, the $B/T$ ratio is in the interval between 2.8 and 3.4 (Figure 18). The results in Figure 22 were obtained for infinite water depth. To obtain a clear validation of the present results, the same experiment is performed for the Lewis ship (Table 1) for different $B/T$ ratios. The results are shown in Figure 13 and show the decay of $c_{22}$ of the ship-like shape versus the $B/T$ ratio. Comparing the range of the $B/T$ ratio and the data from Figure 13, the estimate of the added mass coefficient for the Lewis-type shape lies in the interval $c_{22} \in [0.6, 0.73]$. For each cross-section $k$, the results for the infinitely deep water are shown in Figure 12 for $c_{22}^{(k)}$, $S_k$ and $B/T$ ($\beta_i = 1$). The added mass coefficients of the ship-shaped cross-section are always smaller than the added mass coefficients of the circular cross-section. This fact is mostly related to the $B/T$ ratio.

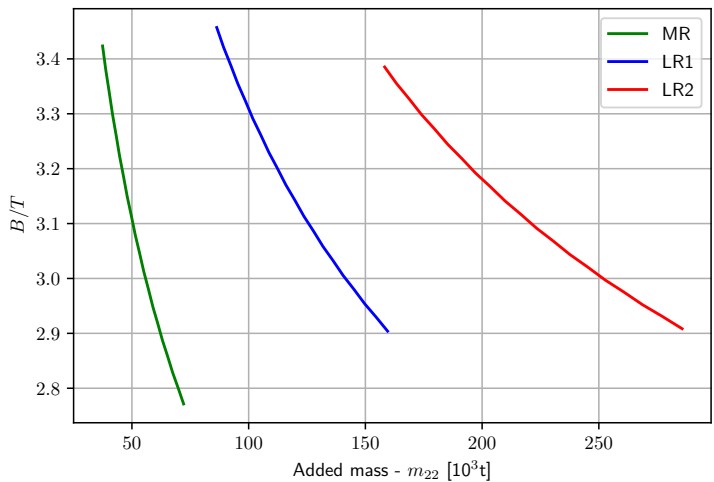

**Figure 18.** Results for the added mass $m_{22}$ of the **MR**, **LR1** and **LR2** tanker type with respect to $B/T$ ratio.

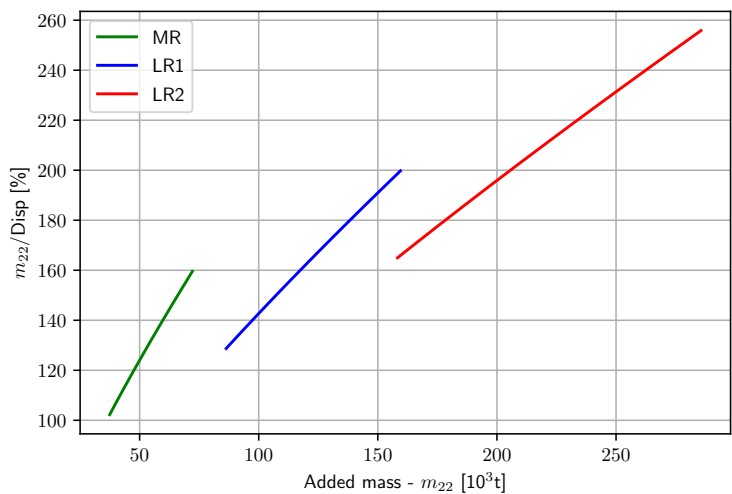

**Figure 19.** Results for the added mass $m_{22}$ of the **MR**, **LR1** and **LR2** tanker type with respect to $m_{22}/\text{Disp}$ ratio.

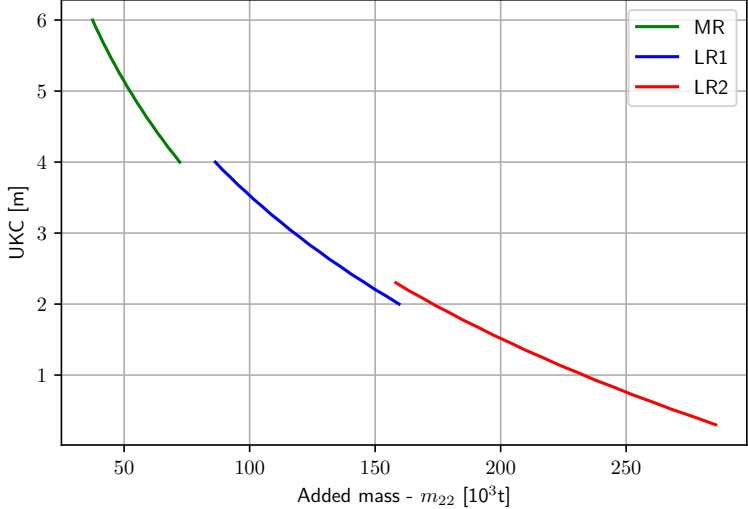

**Figure 20.** Results for the added mass $m_{22}$ of the **MR**, **LR1** and **LR2** tanker type with respect to UKC.

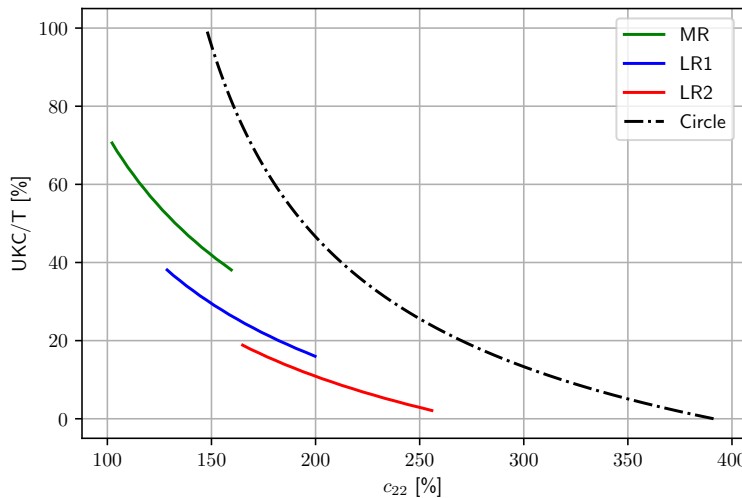

**Figure 21.** Results for the added mass coefficient $c_{22} = m_{22}/\text{Disp}$ of the **MR**, **LR1** and **LR2** tanker type with respect to UKC/$T$ ratio. Dashed line is the same as in Figure 14.

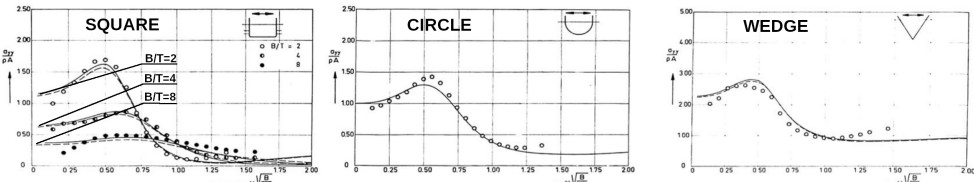

**Figure 22.** Results for the added mass coefficient $c_{22}$ obtained from Vugts [33]. Comparison with present results can be made with the results of zero frequency case $\omega = 0$.

## 4. Discussion and Conclusions

The effect of added mass during the berthing manoeuvre was analysed at the liquid berth in the port of Koper for different types of oil tankers. The formulation of the problem is based on the theory of ideal incompressible fluid so that the velocity of the surrounding fluid can be expressed as a complex velocity potential. Measured ship oscillation times under dolphin loading are long, and the simplification of the zero-frequency limit leads to the simplification of the free surface boundary condition (longwave approximation). The described simplifications and the use of complex analysis methods facilitate the calculation of added mass. One of the missing effects is the viscosity effect. If viscosity were included, it would complicate the system of equations to such an extent that a symbolic solution would not be possible, which was the motivation of this study to avoid numerical calculations as much as possible.

In the present case, the complex velocity potential represents the finite depth situation to include the effect of under keel clearance (UKC) in the calculations of added mass. The simplification of 3D calculations into 2D calculations is applied with the strip theory approach for the zero head velocity. All the described simplifications resulted in a system of equations that can be solved symbolically. The rather complicated system of equations is described in *Python* [41] environment with *SymPy* [42] module for the symbolic calculations and can be found in the *Zenodo* repository [43].

Conformal maps as Lewis map [12] defines a simplified ship geometry with only three parameters. The geometry is simplified, but the overall shape is very close to that of an oil tanker. A similar system is discussed in [34]. The results in [34] are very similar to those in this study for the larger values of UKC/$R$. The sway motion was also analysed in [33] and the results are comparable. The computational system is written in complex Python language form and it is very easy to manipulate with it for a variety of different parameters, cross-section geometries, ship details, UKC etc.

The main objective of this study was to accurately estimate the amount of added mass for certain types of ships docking at the liquid jetty where flexible dolphins are installed. The information of added mass can now be used in future fatigue analyses of flexible dolphins. To support a broader analysis, three different ship types are identified as the representative fleet: MR oil tanker, LR1 oil tanker and LR2 oil tanker. Each class is analysed under different draft conditions with a constant water height of the port basin in the full simulation procedure. In the port of Koper, the average tidal range is about 0.5 m. In this case, the minimum mooring UKC at low tide should always be 10 cm. All these aspects were included in the analyses to obtain accurate data for the ship added mass.

One of the general aspects of added mass in relation to UKC can be reduced from the results shown in Figure 21. With a fair degree of confidence, it can be extrapolated to similar scenarios for different ports and a variety of ships with $Cb \approx 0.8$.

The observed added mass is in the range of 100–160% of displacement for MR oil tanker type, 130–200% of displacement for LR1 oil tanker type and 170–260% of displacement for LR2 oil tanker type. As observed, the values of added mass are very high and must always be considered in the loading analysis of flexible dolphins.

**Funding:** This research was co-funded by ARRS grant number P2-0394.

**Data Availability Statement:** Complete code project available at Zenodo [43].

**Conflicts of Interest:** The author declares no conflict of interest.

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
