# Peer review of "Ships Added Mass Effect on a Flexible Mooring Dolphin in Berthing Manoeuvre"

_jmse, doi:10.3390/jmse9020108_

Round 1

Reviewer 1 Report

1. The descriptions in Figure 5-10 is not enough, though some descriptions has been carried out, yet lacks in proper presentation, it should be compared and discussed in more detail.
2. Legends should be included in the figures (Figure 11-14) though the comparison lines are plot in different colors.

Author Response

Q1. The descriptions in Figure 5-10 is not enough, though some descriptions has been carried out, yet lacks in proper presentation, it should be compared and discussed in more detail.

A1. Enlarged the caption of Figures 5-10 and added an expanded description in the new paragraph (lines 184-192)

Q2. Legends should be included in the figures (Figure 11-14) though the comparison lines are plot in different colors.

A2. Colored text is added to the curves in the caption. In addition to Figures 11-14, new graphs of the same data with labels for different types of ships are added in Figures 18-20 to avoid the possibility of misunderstanding the data.

Reviewer 2 Report

The effect of added mass during the berthing manoeuvre was analyzed at the liquid berth in the port of Koper for different types of oil tankers. This is important because to avoid direct contact with the infrastructure of the liquid cargo terminal, two flexible dolphins reduce the speed of the ship and act like two huge shock absorbers. Today’s cargo terminal was designed for smaller types of ships, but now larger ships arrive in the Port of Koper.   The work developed by the author is of interest, especially the conclusions regarding the added masses in relation to the 3 sizes of vessels studied. However, I have a series of questions to address to the author:   -Applicability of the results to the reality of other ports. Possibility of adapting the conclusions to this generality.   - It is clear in the case of class MR the correspondence with fig. 12, but I would like to know if the added mass for vessels LR1 and LR2 corresponds to the content of figures 13 and 14.   - Explanation of the 10 cm UKC condition. it is surprising

Author Response

Q1. Applicability of the results to the reality of other ports. Possibility of adapting the conclusions to this generality.

A1. New diagrams are added in the results, in particular Figure 21 can be used as a generalisation of the data to other ports for ships with Cb=0.8. A discussion of the generalisation can be found in the paragraph between lines 256-270.

Q2. It is clear in the case of class MR the correspondence with fig. 12, but I would like to know if the added mass for vessels LR1 and LR2 corresponds to the content of figures 13 and 14.

A2. There is a detailed validation of the results with the relevant literature, the solution for infinite depth and the analytical solution for circular gemoetry. The data in Figures 13 and 14 now appear to be good and in agreement with the possible validation. The description of results in lines 242-270 attempts to verify the validation procedure.

Q3. Explanation of the 10 cm UKC condition. it is surprising

A3. It can be related to A.2

Reviewer 3 Report

The literature review is very few and the author did not clearly describe the contents of the earlier studies. The author also did not describe how the results were obtained and no validations were included either. The paper is not accepted at the current form and major revision is needed.

Author Response

Q1. The literature review is very few and the author did not clearly describe the contents of the earlier studies.

A1. Major revision of "Introduction" section with literature review of older studies is done in lines 35-74. Relevant references from related field added.

Q2. The author also did not describe how the results were obtained and no validations were included either.

A2. In the uploaded version, the "Results" section has been completely revised. All the steps of how the results were determined are explained. There is a detailed validation of the results with the relevant literature, the solution for infinite depth and the analytical solution for circular geometry. The description of the results in lines 242-270 attempts to verify the validation procedure.

Q3. The paper is not accepted at the current form and major revision is needed.

A3. The newly uploaded paper has undergone a thorough revision. The Introduction and the Results section have been completely reformulated. Also, new experiments were included for the validation procedure: Solution for indefinite depth (Figure 13), Vughts results (Figure 22) and the circular section added mass coefficient with respect to UKC (Figure 21). The new validation procedure attempts to verify the results.

Round 2

Reviewer 2 Report

  The author has answered the questions appropriately and made the proposed changes, for this reason the article is accepted for publication

Reviewer 3 Report

The author has properly amended the manuscript according to the comments given. Although the fluid viscosity is neglected, the results are still worth for the validation of later numerical simulations made by other researchers.